

# Combining occurrence and abundance distribution models for the conservation of the Great Bustard

Chunrong Mi[1], Falk Huettmann[2], Rui Sun[3] and Yumin Guo[1]

[1] College of Nature Conservation, Beijing Forestry University, Beijing, China
[2] EWHALE Lab, Department of Biology and Wildlife, Institute of Arctic Biology, University of Alaska—Fairbanks, Fairbanks, AK, United States of America
[3] Key Laboratory of Water Cycle and Related Land Surface Processes, Institute of Geographic Sciences and Natural Resources Research, University of Chinese Academy of Sciences, Beijing, China

## ABSTRACT

Species distribution models (SDMs) have become important and essential tools in conservation and management. However, SDMs built with count data, referred to as species abundance models (SAMs), are still less commonly used to date, but increasingly receiving attention. Species occurrence and abundance do not frequently display similar patterns, and often they are not even well correlated. Therefore, only using information based on SDMs or SAMs leads to an insufficient or misleading conservation efforts. How to combine information from SDMs and SAMs and how to apply the combined information to achieve unified conservation remains a challenge. In this study, we introduce and propose a priority protection index (PI). The PI combines the prediction results of the occurrence and abundance models. As a case study, we used the best-available presence and count records for an endangered farmland species, the Great Bustard (*Otis tarda dybowskii*), in Bohai Bay, China. We then applied the Random Forest algorithm (Salford Systems Ltd. Implementation) with eleven predictor variables to forecast the spatial occurrence as well as the abundance distribution. The results show that the occurrence model had a decent performance (ROC: 0.77) and the abundance model had a RMSE of 26.54. It is noteworthy that environmental variables influenced bustard occurrence and abundance differently. The area of farmland, and the distance to residential areas were the top important variables influencing bustard occurrence. While the distance to national roads and to expressways were the most important influencing abundance. In addition, the occurrence and abundance models displayed different spatial distribution patterns. The regions with a high index of occurrence were concentrated in the south-central part of the study area; and the abundance distribution showed high populations occurrence in the central and northwestern parts of the study area. However, combining occurrence and abundance indices to produce a priority protection index (PI) to be used for conservation could guide the protection of the areas with high occurrence and high abundance (e.g., in Strategic Conservation Planning). Due to the widespread use of SDMs and the easy subsequent employment of SAMs, these findings have a wide relevance and applicability than just those only based on SDMs or SAMs. We promote and strongly encourage researchers to further test, apply and update the priority protection index (PI) elsewhere to explore the generality of these findings and methods that are now readily available.

Corresponding author
Yumin Guo, guoyumin@bjfu.edu.cn

## INTRODUCTION

The knowledge of species occurrence and abundance distribution provides fundamental information for conservation biology (*VanDerWal et al., 2009*; *Drew, Wiersma & Huettmann, 2011*; *Primack, 2012*; *Johnston et al., 2015*). Understanding how environmental factors are related to species occurrence and abundance distribution and how they are explicit in time and space are priorities in current biodiversity conservation (*Drew, Wiersma & Huettmann, 2011*; *Martín et al., 2012*).

Species distribution models (SDMs) are empirical ecological models that relate species observations to environmental predictors (*Guisan & Zimmermann, 2000*); usually this process is done using machine learning algorithms (*Drew, Wiersma & Huettmann, 2011*, see *Mi et al., 2017* for an application). SDMs have become important and essential tools in ecology, biogeography, climate change research, conservation, and management because of on their spatial occurrence prediction capacities (*Peterson et al., 2002*; *Guisan & Thuiller, 2005*; *Elith et al., 2006*; *Araújo & New, 2007*; *Mi, Huettmann & Guo, 2016*). SDMs built with count data are called species abundance models (SAMs) (*Elith & Leathwick, 2009*; *Barker, Cumming & Darveau, 2014*; see *Yen, Huettmann & Cooke, 2004* for an application). SAMs are still less commonly used, despite providing valuable information for conservation and management. However, increasing attention has been paid to these problems in recent years (e.g., *Yen, Huettmann & Cooke, 2004*; *Martín et al., 2012*; *Howard et al., 2015*; *Ashcroft et al., 2017*; *Fox et al., 2017*).

In the past, spatial conservation decisions and plans were usually just based on SDMs (e.g., *Suárez-Seoane et al., 2008*; *Gray et al., 2009*; *Adams et al., 2016*; *Mi, Huettmann & Guo, 2016*). However, despite statements by *Newton (2008)*, many scholars found that species occurrence and abundance distribution did not to display similar patterns (*Yen, Huettmann & Cooke, 2004*; *Karlson, Connolly & Hughes, 2011*; *Yin & He, 2014*; *Johnston et al., 2015*). The difference may represent a mixture of effects and may reflect the differences between the underlying biological processes of abundance and occurrence (*Johnston et al., 2015*). Therefore, conservation decisions only based on SDMs predictions are insufficient and may even be misleading; the same applies for SAMs. In the future, one time-critical challenge and associated progress will be centered how to combine the useful information that SDMs and SAMs each offer for conservation.

In this study, we evaluated a case study using the endangered Great bustard (*Otis tarda dybowskii*), which winters in Cangzhou in the North China Plain near Bohai Bay. This area is one of the most important wintering grounds for this species (approximately 300 individuals, c.13.6∼20.0% of China's total wintering population (*Goroshko, 2010*; *Meng, 2010*)). Using the Great Bustard as a case study would contribute to our conservation knowledge about the habitat use of this threatened species and enable us to design better conservation policies. By studying not only the spatial occurrence and the abundance

patterns but also combining these two model types together as a role model, predictive modeling and its inferences would potentially have wider conservation implications. Our overall objective of this research was to: (1) assess and develop models to accurately predict the patterns of bustard occurrence and abundance; (2) identify the environmental variables that influence the occurrence and abundance of this species; (3) combine occurrence and abundance models as a new contribution to conservation decisions; and (4) investigate the overall relationship among predicted occurrence, predicted abundance and observed abundance. Well-tested and suitable methods used in this research could be useful for the conservation of the Great Bustard, and other rare species; additionally, this research could generally improve biodiversity through the application of SDMs and SAMs.

## MATERIALS AND METHODS

### Study area

This study was conducted at the wintering ground of the endangered Great Bustards in Cangzhou, southeast of the Heibei Province in the wider Bohai Bay (Fig. 1). It is located at $38°12'57''$–$38°36'51''$ latitude and at $116°50'48''$–$117°24'03''$ longitude in the warm temperate, semi-humid monsoon climate zone, which features the slightly marine climatic characteristic of the Bohai Sea region. The topographical and climate conditions vary little in the study area (altitude varies by 13 m, temperature by 0.4 °C, and precipitation is the same). The total study area is 2,191.4 km$^2$, consisting of farmland (1,675.1 km$^2$; 76.4%), residential area (330.5 km$^2$, 15.1%), open water (23.5 km$^2$; 1.1%) and other unspecified land uses (e.g., home lots, sheds).

Most of the farms in this region produce cereal, which is grown in a 2-year rotation system. In the first year, winter cereal is cultivated from early September to the end of April of the following year. Then, corn is cultivated between the end of April to early September of the same year. The study area was chosen (Fig. 1) because of its large number and proportion (approximately 300 individuals, c.13.6 ~20.0% of China's total wintering Great Bustard population (*Goroshko, 2010*; *Meng, 2010*)). This area is the world's largest wintering ground of the endangered *O. t. dybowskii*. This area is representative of the typical farmland in the North China Plain. In addition, accurate Great Bustard census data, geographic information system (GIS) data coverages and satellite imagery were readily available.

### Bird census data

Spatial occurrence and abundance data for Great Bustards were used to develop models. A Great Bustard census was conducted between November 2013 and March 2014. In the study area, we travelled with a small four-wheel-drive tractor along the roads between farmland, at speeds between 10–30 km/h. We are confident we have an equal and virtually complete detectability in the study area. No great bustard flocks were overlooked. Our team consisted of two experienced observers (one surveyor and one local resident) carrying out the bustard counts, and this team was familiar with the survey area. When a flock was found, we drove slowly and stopped at a location approximately 100–500 m distance from the bustard flocks; then, we recorded the size, location, habitat type and basic behavior of
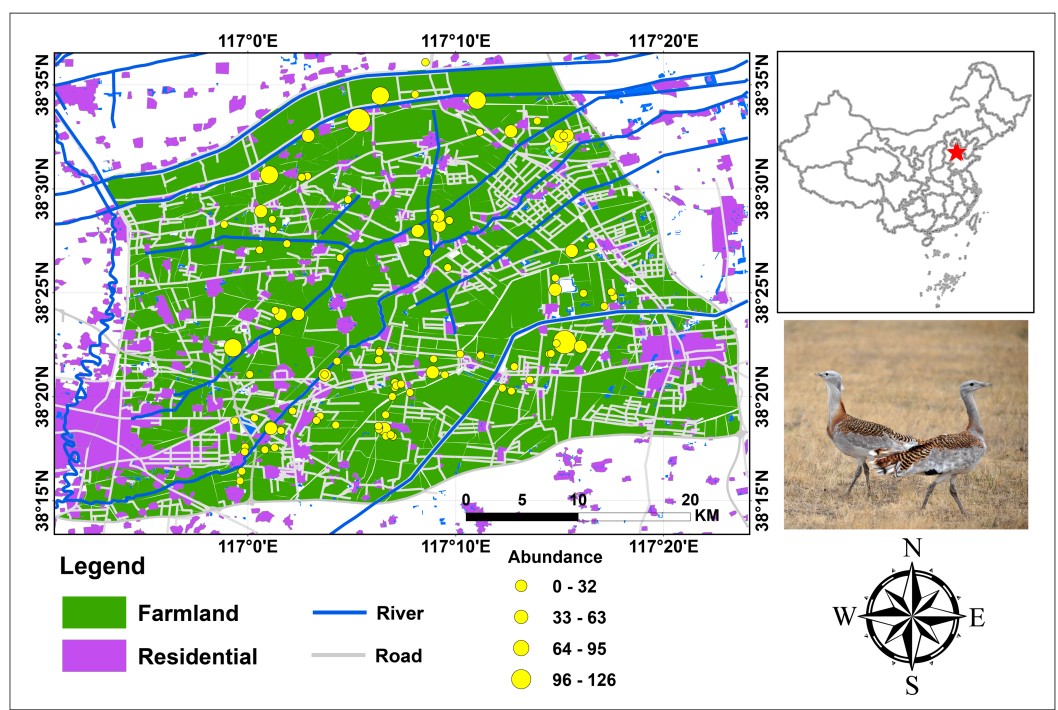

**Figure 1** **Study area and bird abundance and occurrence data for Great Bustard in Cangzhou, China.** Photograph of Great Bustard by Jianguo Fu.

the flock. This resulted in the high detection of birds and flocks in the study area because birds can be seen from long distances (∼3 km) and also when flying away. The actual coordinates of the animal locations were obtained by Google Earth by combining it with our recorded locations. Each census was conducted from dawn until dusk. During the study, we identified 94 bustard sites within the study area. To our knowledge, this census data comprises the best available data for bustards in China.

## GIS environmental layers

Based on the environmental conditions in our study area, we selected eleven habitat and landscape (i.e., environmental) variables to construct models that predict occurrence and abundance (Table 1). To obtain these variables, we acquired the basemap from Google Earth (using Daogle, an open source software made by a Chinese individual: http://www.daogle.com/; as used and explained in *Mi, Huettmann & Guo, 2014*) and derived otherwise unavailable high-resolution landscape inventory information about open-water pools, rivers, residential areas, national roads, provincial roads, expressway, farmland roads, ditches and farmland areas from the base map. Next, we constructed a distance layer for these variables (except for the farmland area) using the Euclidean Distance tool in ArcGIS 10.1 (ESRI, Redlands, WA, USA) with a 30 m × 30 m pixel size (ArcToolBox-Spatial Analyst Tools-Distance-Euclidean Distance). This high-pixel resolution was chosen to maintain consistency with the remote sensing variable resolution we used.

**Table 1 Comparison of features around 94 sites occupied by great bustards and 10,000 random points.** Values are means ± standard deviations.

| Layer | Variable | Description | Bustard sites | Random points |
|---|---|---|---|---|
| 1 | Distance to pool | Distance to pool in meter | 1,179.0 ± 734.5 | 1,378.0 ± 910.3 |
| 2 | Distance to river | Distance to river in meter | 2,302.0 ± 1,751.2 | 2,630.0 ± 2,483.0 |
| 3 | Distance to residential | Distance to residential in meter | 935.0 ± 586.8 | 980.2 ± 723.8 |
| 4 | Distance to national road | Distance to national road in meter | 5,280.0 ± 4,234.2 | 5,855.0 ± 4,036.9 |
| 5 | Distance to provincial road | Distance to provincial road in meter | 8,730.0 ± 5,928.7 | 9,217.0 ± 6,112.4 |
| 6 | Distance to expressway | Distance to expressway in meter | 10,010 ± 5,750.0 | 9,585.0 ± 6,666.7 |
| 7 | Distance to farmland road | Distance to farmland road in meter | 477.4 ± 385.3 | 524.9 ± 455.8 |
| 8 | Distance to ditch | Distance to ditch in meter | 1,522.0 ± 1,722.7 | 2,120.0 ± 2,078.1 |
| 9 | Area of farmland | Area of farmland in kilometers | 3.3 ± 3.2 | 5.3 ± 6.2 |
| 10 | MNNDVI | The average value of the normalized difference vegetation index from November, 2013 to March, 2014 | 0.14 ± 0.04 | 0.13 ± 0.05 |
| 11 | MAXNDVI | The maximum value of the normalized difference vegetation index from November, 2013 to March, 2014 | 0.23 ± 0.06 | 0.21 ± 0.07 |

## Satellite images

A range of the best cloud-free HJ-1A/B (HuanJing (HJ)) satellite images (http://www.cresda.com) with 30 m × 30 m resolution were obtained for each month between November 2013 and March 2014 in order to calculate the normalized difference vegetation indices (NDVI) signature for each pixel. The HJ-1A/B CCD data were run for radiometric calibration, atmospheric correction and geometric correction to obtain surface reflectance data and subsequent NDVI data. Radiometric calibration was finished using 2014 HJ-1A/B CCD absolute radiometric calibration coefficients, which were provided by the China Center for Resources Satellite Data and Application. For this study, we used maximum and mean NDVI to represent the vegetation conditions (*Osborne, Alonso & Bryant, 2001*).

## Model development

We employed a machine learning technique, Random Forest, to model the occurrence as well as the abundance distribution of Great Bustards. *Breiman*'s (*2001*) Random Forest implementation in SPM7 by Salford Systems Ltd. (San Diego, CA, USA) is robust to over-fitting and is widely recognized to produce high-quality predictive models (*Mi et al., 2017*). Hence, Random Forest is increasingly applied to species distribution modelling (*Cutler et al., 2007*; *Drew, Wiersma & Huettmann, 2011*; *Mi, Huettmann & Guo, 2016* for an application using bustards in China). Though Random Forest performed the best in terms of predicting abundance itself (see Appendix S1), testing the feasibility of other data was essential for maintaining high certainty. Thus, to assess the robustness of the model, we pooled data from 2013 and 2014, and then used 80% of the abundance data as training data and the remaining 20% as testing data. When we constructed initial abundance models with all eleven environmental predictors, model performance was not good (the $R^2$ value was small). This was likely due to the regression settings in the Random Forest algorithm. For a better outcome, we assessed a ''stepwise'' setting in SPM
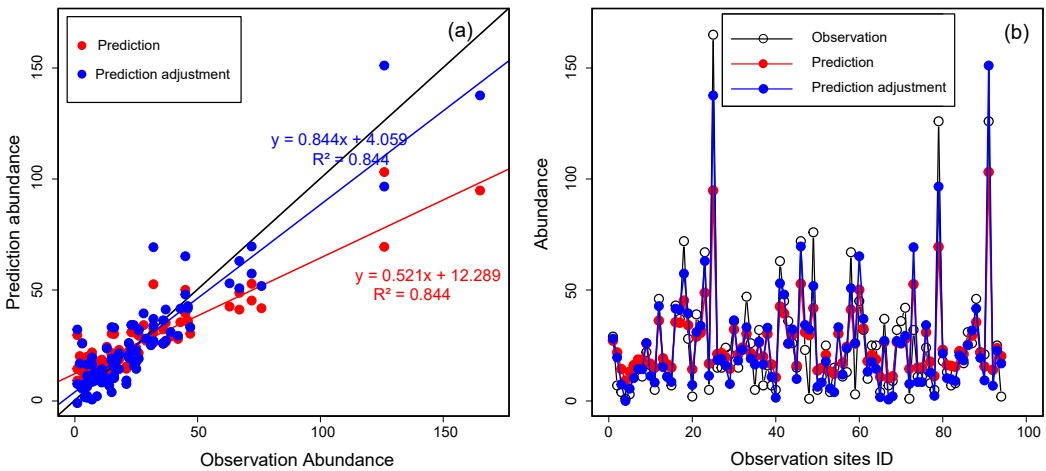

**Figure 2** **The relationship between observation and prediction abundance using Random Forest for Great Bustards.** (A) Scatter plot of observation abundance with prediction and adjustment prediction abundance. The black line is the expected 1:1 relationship. (B) Lines and points plot of observation, prediction and adjustment prediction abundance.

for all of the abundance data (100%), re-run the models, and obtained better results. As a result, we identified a multivariate set of four environmental predictors (distance to expressway, distance to national road, distance to pool, and MNNDVI), which had the best performance (the biggest $R^2$ value). Using these four predictors, we reconstructed the abundance model based on the training data (80%) and validated it with testing data (20%). We found that the regression model performance was acceptable but fair ($R^2 = 0.551$) when comparing observed abundance with predicted abundance. Thus, we constructed the final abundance model based on the above four selected variables described above and the entire set of observation data. To obtain an abundance index more closely aligned with the observations, we adjusted the prediction abundance according to the linear regression between prediction abundance and observation. First, we constructed a regression formula based on the known abundance from observation and prediction [observation abundance = A × prediction abundance + B]. Then, once A and B were be known, they were applied to calculate the adjustment abundance = [adjustment abundance = A × prediction abundance + B]. The regression relation between observation and prediction abundance are shown in Fig. 2.

Further, Random Forest was also applied to rank the relative importance of the environmental variables. In SPMv7, partial dependence plots are not directly implemented in Random Forest but can be obtained easily in R or can be mimicked in the TreeNet model (*Friedman, 2002*) as a Random Forest run. Thus, we used TreeNet with bagging settings to create partial dependence plots for each variable included in the occurrence and abundance models.

Approximately 10,000 pseudo-absence points were chosen by random sampling across the study areas using the freely available Geospatial Modeling Environment (GME) software (http://www.spatialecology.com/gme/) for distribution models. In SPM7, we set balanced

class weights, grew each model to 1,000 classification trees for the occurrence model and 1,000 regression trees for the abundance model, and used all other default setting of the software. We extracted the habitat information for presence and pseudo-absence points for Great Bustards from the environmental layers in GME ("isectpntrst" commands), and then we created a model file in SPM7 called a 'grove' that contained the algorithm that quantified the patterns of occurrence; this was used to score all pixels in the study area. We also extracted the habitat information from the same environmental layers for abundance points and then generated a 'grove' file for abundance to score abundance estimates for each pixel in the study area.

For spatial occurrence and abundance distribution visualization, we applied the SPM7 grove files to a regular lattice of points (pixels; also attributed to the environmental variables) spaced at 30-m intervals across the study area. Model outputs generated relative indices of occurrence (RIO; an index of pixels from 0 to 1 that represent a relative index belonging to the 'occurrence' class) and a relative abundance index (prediction abundance) for each point in the regular lattice based on its underlying environmental variables. We also adjusted the predicted abundance based on a linear regression as constructed in the previous model development steps (Fig. 2A). For a better continuous spatial visualization, the RIO and predicted abundance values were smoothed between neighboring points across the extent of the study area using the Inverse Distance Weighting (IDW) tool in ArcGIS 10.1. This yielded spatially continuous predictive distribution and abundance raster maps of for the Great Bustard.

## Model validation

The Random Forest performance was first assessed internally using a set of 'out-of-bag' (OOB) training points (OOB; a specific concept used with Random Forest models to describe a subset of points not used initially for model fitting; *Breiman, 1996*; *Breiman, 2001*). Using this out-of-bag dataset, the receiver operating characteristic (ROC) and RMSE were used to calculate the predictive performance of the occurrence and abundance models, respectively (*Zweig & Campbell, 1993*; *Fielding & Bell, 1997*; *Huettmann & Gottschalk, 2011*).

## Priority protection analysis

To have a more suitable and scientific protection plan for the endangered Great Bustard, in this study, we developed and proposed the use of an index called the priority protection index (PI), which combines the predicted results of the SDM and SAM. This index is calculated by the following equation for each site:

$$PI = \frac{RIO \times RA}{\max(RIO \times RA)} \tag{1}$$

where $PI$ = priority protection index (an index of pixels from 0 to 1 that represent the priority of conservation), $RIO$ = relative index of occurrence, and $RA$ = relative abundance (prediction abundance). In our study, we computed the PI for the entire study area based on the RIO and the adjusted RA value for each grid cell of the spatial occurrence and abundance maps. Then, we used the IDW tool in ArcGIS 10.1 to generate spatially continuous priority protection index (PI) raster maps. In this equation, we did not consider the weighting the

**Table 2** Variables importance ranking of occurrence and abundance models.

| Ranking | Occurrence model | Abundance model |
| --- | --- | --- |
| 1 | Area of farmland | Distance to national road |
| 2 | Distance to residential | Distance to expressway |
| 3 | Distance to ditch | Distance to pool |
| 4 | Distance to expressway | MNNDVI |
| 5 | Distance to pool | – |
| 6 | Distance to river | – |
| 7 | Distance to provincial road | – |
| 8 | Distance to national road | – |
| 9 | Distance to farmland road | – |
| 10 | MAXNDVI | – |
| 11 | MNNDVI | – |

biotic and socioeconomic variables. Therefore, the justification and use of the PI should be explained a little more: when combining the SDM with the SAM, one will not find a straight forward relationship between occurrence and abundance (see *Yen, Huettmann & Cooke, 2004* for an example). What the PI will do, but what has not been achieved much before, is to essentially model the relationship between occurrence and abundance, and provide a combined view of the occurrence index and abundance index that is explicit in space and time. Achieving this can help to better prioritize the pixels.

## RESULTS

### Model performance

Our distribution model obtained a decent performance (ROC: 0.77) according to *Fielding & Bell (1997)*, and the abundance model had a RMSE of 26.54 (RMSE is unit-less).

### Variable importance

Table 2 presents the variable importance ranking for the occurrence and abundance models obtained from the Random Forest method. We found that the area of farmland, the distance to residential areas (buildings), to ditches and to expressways were the top four most important variables that influenced bustard occurrence. Those come as a multivariate ecological package (a combination of many predictors). The NDVI, which represents vegetation conditions, was less important than the other nine predictors. For the abundance model, the most important factors were the distance to national roads and to expressways, followed by water factors (the distance to pools) and food-related factors (MNNDVI).

### Partial dependence plots

Partial dependence plots could interpret the functional relationships and effects of each variable by representing a variable's marginal effects on the response (*Elith, Leathwick & Hastie, 2008*; *Johnstone et al., 2010*). It helps to find the signal in the data; Fig. 3A indicates that the occurrence preference of bustards for farmland areas was between 0.6 and 7.5 km$^2$.

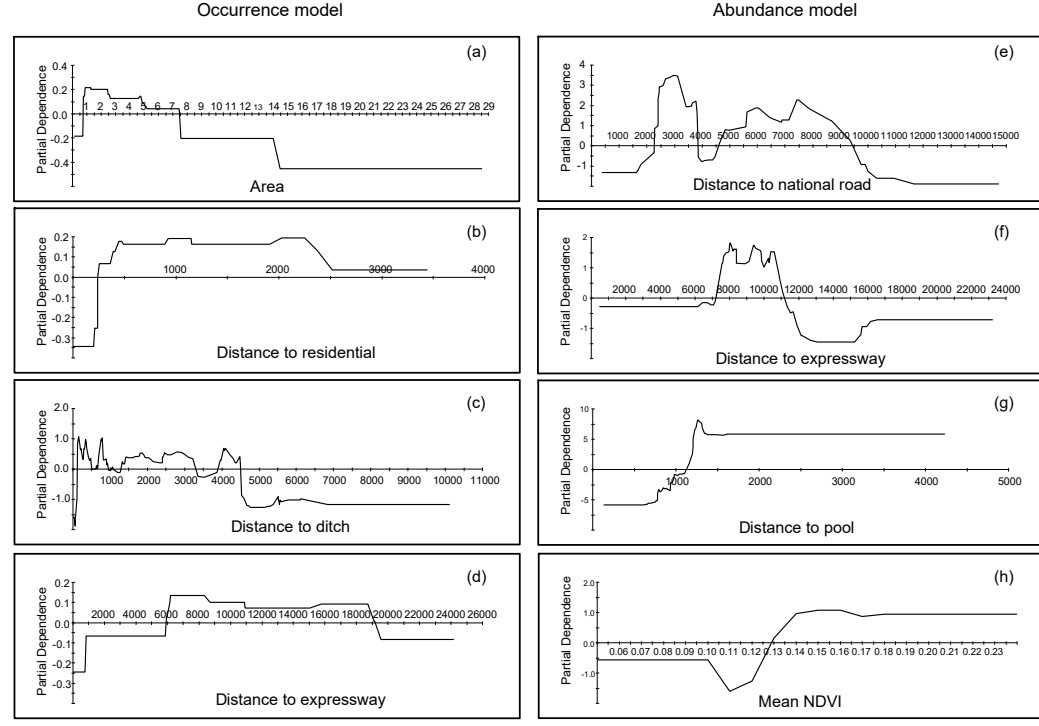

**Figure 3 Partial dependence plots for the top four most influential variables in the occurrence and abundance distribution models for Great Bustards, respectively.** (A) Area of farmland in occurrence distribution model; (B) distance to residential in occurrence distribution model; (C) distance to ditch in occurrence distribution model; (D) distance to expressway in occurrence distribution model; (E) distance to national road in abundance distribution model; (F) distance to expressway in abundance distribution model; (G) distance to pool in abundance distribution model; and (H) mean NDVI in abundance distribution model.

Additionally, based on model results, it appeared the bustard preference were as follows: distance to residential areas ranged from 250 to 2,500 m (Fig. 3B), distance to ditches ranged from 100 to 4,500 m (Fig. 3C), and distance to expressways ranged from 6,000 to 19,000 m (Fig. 3D). In contrast, for abundances, more individuals occurred beyond 2,300 m, but were less than 9,500 m away from national roads (Fig. 3E); additionally, bustards were found in a range between 7,000 and 11,000 m away from expressways (Fig. 3F). Moreover, this species stayed away from pools (maintaining distance greater than 1,500 m, Fig. 3G) and preferred areas with more vegetation (mean NDVI during the investigation was larger than 0.13, Fig. 3H). The information for other variables, which were more marginal, can be found in Appendix S2.

## Occurrence, abundance distribution patterns and priority protection

Figure 4 shows the maps of the RIO (relative index of occurrence), adjusted RA (relative abundance) and PI (priority protection index). From the RIO map (Fig. 4A), we found that the distribution area of high RIO for bustards was high. The regions of high possibility of bustard occurrence were concentrated in the south-central study area; and the whole

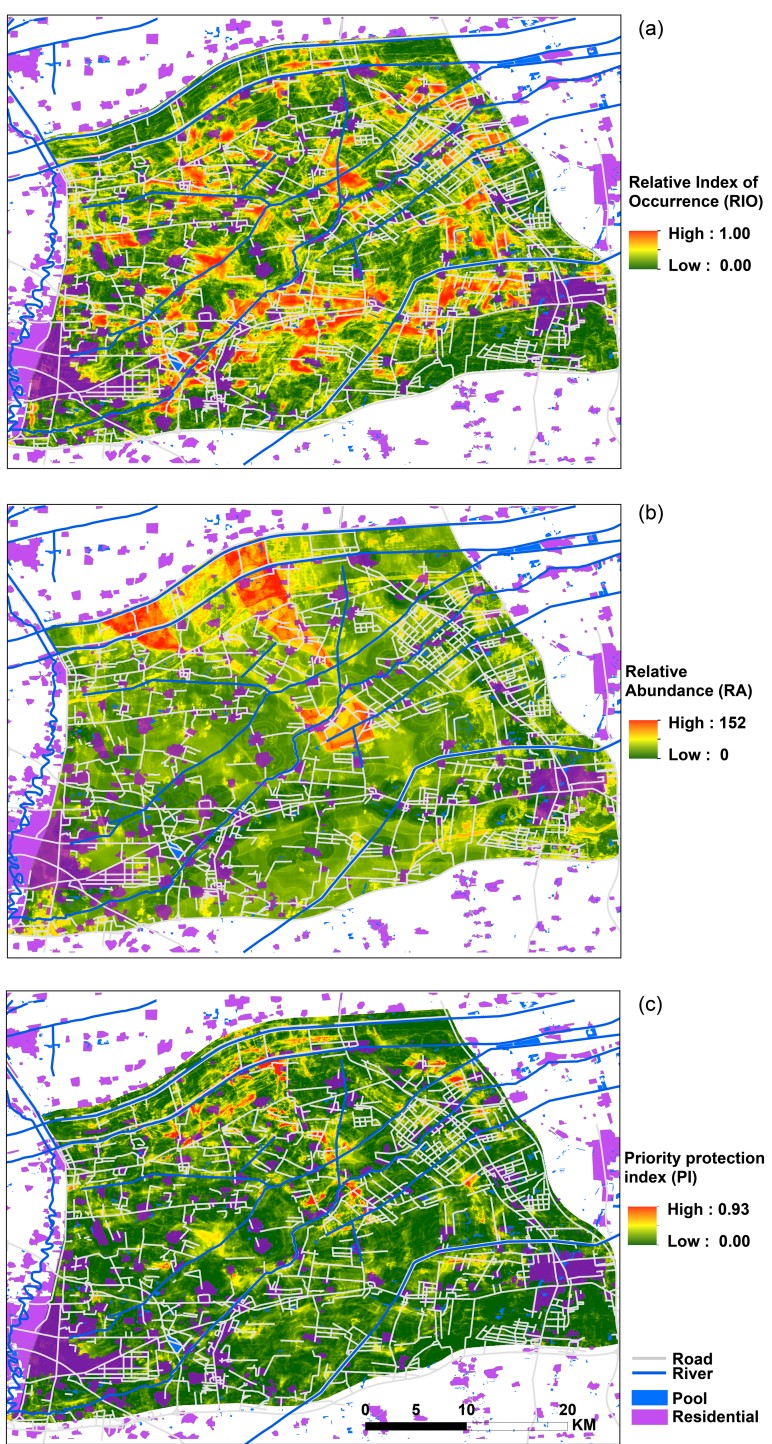

**Figure 4** **Spatial distribution map of relative index of occurrence (RIO), relative abundance (RA) and priority protection index (PI).** (A) Map of relative index of occurrence (RIO); (B) map of adjusted relative abundance (RA); and (C) map of priority protection index (PI).

habitats represented a fragmented distribution. The abundance distribution had a different pattern and showed high populations occurrence in the central and northwestern parts of the study area (Fig. 4B). Based on the occurrence and abundance distribution results, we used Eq. (1) and obtained the results shown in Fig. 4C. These results indicate that a high PI is located in the center, north and northeast regions of the study area, and they indicate a sporadic and fragmented distribution, which could represent a priority protection site if a conservation decision was to be made.

## DISCUSSION

The occurrence and abundance models of Great Bustard developed here were designed to identify relevant locations for where to prioritize conservation, and to assess the effects of each variable that influenced this species' occurrence and abundance (Fig. 3). From a multivariate perspective, the area of farmland, distance to residential areas, distance to ditches and to expressways were among the top four most important predictors for bustard occurrence; in contrast, for the abundance model, the variables consisted of another multivariate package that include the distance to national roads, the distance to expressways, the distance to pools and the mean NDVI (Table 2). We found that high RIO habitats had a fragmented distribution throughout the entire study area (Fig. 4A). The abundance model showed that large populations usually occurred in the central and northwestern parts of our study area (Fig. 4B). The center, north and northeast portions of the study area had a high priority protection index (PI) and had a severely fragmented distribution, including these areas should be the priority sites for protection (Fig. 4C). This not only confirms our own records and, with the help of the PI, can now be quantified and modeled further for more effective conservation application.

In our study area, human disturbance was very strong and represented by indicators, such as density of roads and residential areas (Fig. 1). During our study, we also found other threats to this endangered species: these included farmers grazing their sheep; famers sprinkling poison baits in the wheat fields to present sheep from entering; some bird photographers pursued bustards by walking or following birds while on motor vehicles to take photos, which they wanted to show off to others; hunters with dogs chasing hare and ring-necked pheasant during the day and night; some local people hunting bustards; increasing power lines construction in agricultural lands, resulting in bustards sometimes colliding with wires, getting injured, or even dying, especially flying on foggy days or when in a hurry (*Janss & Ferrer, 2000*); and the interference of firecracker sounds during the Chinese Spring Festival as well as oil rigs and wind farms. Though carrying a high disturbance can result in stress synthesis (e.g., "death by thousand cuts"), a large number of wintering bustards (approximately 300, c. 13.6 ~20.0% of China's total wintering population; *Goroshko, 2010*; *Meng, 2010*) still wintered in this area. In times of climate change, it can be assumed the population widens (*Mi, Huettmann & Guo, 2016*). Thus, this is an area of essential importance for bustards in China, regardless of which perspective is taken. Therefore, a feasible conservation plan should be designed, and based on our model's prediction result, combined with local public customs and financial support as

well as a wider buy-in. In our opinion, improving the local pupulation's education on animal protection, as has been done over the years, would be useful. The same applies to increasing budgets, enforcement and frequency of patrol by the local management and conservation NGOs in the regions with high PI value, and the local community and government should provide more financial support. However, when designing patrol routes designation in the field, the individuals who are monitoring should avoid getting too close to bustards, so as not to disturb and stress the regular wintering activities of the bustards. For the benefit of this species and its habitats, we suggest not converting crop farmland into nursery farmland; we also encourage farmers to harvest their crops with a machine, which is a more beneficial harvesting method for bustards based on our previous research results (*Mi, Huettmann & Guo, 2014*). We also highly recommend, if possible, to bury power lines into the ground and to collect hunting guns from the local public.

In this study, occurrence and abundance did not display identical spatial distribution patterns, a result which has been reported in some previous studies (*Conlisk, Conlisk & Harte, 2007*; *Karlson, Connolly & Hughes, 2011*; *Yin & He, 2014*; *Johnston et al., 2015*). There is actually no reason to assume a presence site represents only one individual animal, nor should a linear relationship between RIO and abundance be assumed. Technically speaking, 'presence' can mean one to infinite animals are present, and details depend on the actual pixel setup and how it fits into the obtained model. Therefore, while the relationship is not automatically clear, this could be due to several reasons and depending on specific habitat details: Firstly, the environmental variables that contributed to occurrence and abundance were different, as indicated in Table 2. Secondly, the predictors of preference for bustard occurrence and abundance models were different. For instance, bustards occurred in areas with a distance to expressways ranging from 6,000 to 19,000 m (Fig. 3D), while most populations occurred between 7,000 and 11,000 m from expressways in terms of abundance (Fig. 3F) (see more details in Fig. 3 and Appendix S2). Thirdly, they differed in their spatial distribution for occurrence and abundance (Figs. 4A and 4B). Based on the analysis of overlaying the observation sites with the RIO and observation abundance (Figs. 5A and 5B), the estimated relative index of occurrence (RIO) was not consistently related with the relative index of abundance (Fig. 5A). All locations of observed abundance had high RIO (Fig. 5A), and the relationships between occurrence and abundance estimates were nonlinear (Fig. 5B). These differences may represent a mixture of effects that reflect differences between the underlying biological processes that give rise to specific abundance and occurrence at a specific pixel, as well as limitations imposed by the data and methodology used to estimate these patterns (*Johnston et al., 2015*; see *Buckland et al., 2016* for Distance Sampling and detection problems). In addition, how to interpret the inconsistency between these two indices of plant prediction is a problem waiting to be further resolved e.g., between crop occurrence index (equal to habitat suitability index) and crop abundance (e.g., production).

Treating all presence reports as equal in species distribution models (SDMs:occurrence model, habitat niche model) regardless of the abundance of individuals that the habitat supports could provide us with the information on the loss of habitat suitability (*Howard et al., 2014*). Applying models based on abundance data, even at a relatively coarse scale, can

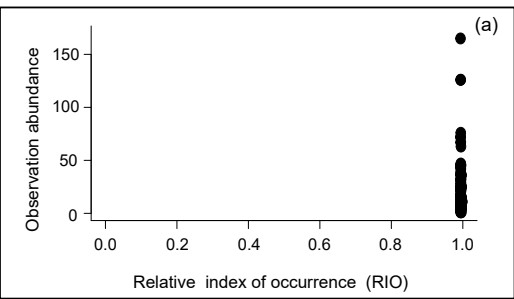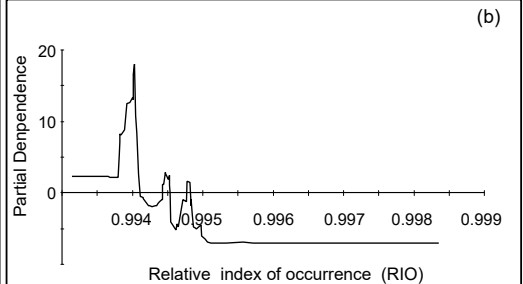

**Figure 5** **Plots of the relationship between relative index of occurrence (RIO) and observation abundance.** (A) Scatter plot between relative index of occurrence (RIO) and observation abundance; and (B) partial dependence plot between relative index of occurrence (RIO) and observation abundance (obtained from TreeNet, non-parametric method).

help to predict spatial patterns of occurrence that are modelled with even greater refinement (*Howard et al., 2014*). Conservation decision-making should use as much knowledge and information as possible to optimize the benefits of conservation actions (*Sutherland et al., 2004*; *Segan et al., 2011*). The use of species distribution models (SDMs) of occurrence has been an important tool in optimizing the selection of protected areas (*Franklin, 2013*; *Guisan et al., 2013*; *Mi, Huettmann & Guo, 2016*; *Han et al., 2017*) based on the ecological niche space (*Drew, Wiersma & Huettmann, 2011*). However, relative abundance is often perceived to be a more relevant metric because it can quantify animals within a pixel, and thus, populations (*Johnston et al., 2015*). Modeling abundance requires methods that can handle large numbers of zero counts as well as the rare, but important, high counts (*Welsh et al., 1996*), even without a solid research design, according to frequentist statistics. However, *Yen, Huettmann & Cooke (2004)*, *Magness, Huettmann & Morton (2008)* and *Fox et al. (2017)* have already shown how machine learning can change this perspective and provide very powerful solutions.

High counts and their locations are particularly important because the pixels with the highest densities of animals are potentially of the greatest interest for conservation planning (*Johnston et al., 2015*). In our study, we found that the regressions in Random Forest performed poorly at sites with low and high counts (Fig. 2B), although it showed a highly linear relationship between observed and predicted abundance ($R^2 = 0.844$; Fig. 2A). Therefore, we argue that the regression method in the Random Forest algorithm should optimize low- and high-count predictions. We recommend to classifying abundances into bins (e.g., high, medium, and low with associated abundance estimates) because Random Forest is exceptionally strong for classification problems. However, for now, this remains an open field of research, but we find our progress remains substantial.

Abundance data could also provide valuable baselines against which to assess future changes (*Cumming, 2007*) (e.g., climate change, land use change). Such changes in abundance will be much more rapidly apparent, and hence, more rapidly detected than changes in presence-absence patterns across large ranges (*Gregory et al., 2005*). However, only a few spatial distribution modelers derived models based on the collection

of abundance data (e.g., *Yen, Huettmann & Cooke, 2004*; *Fox et al., 2017*). This may be because collection of abundance data is more cost or resource demanding than collecting presence—absence data especially for highly mobile animals. Such data are sophisticated in structure and research design, and still they are rarely shared (see in http://GBIF.org). Therefore, we recommend that abundance data could be collected (and can also easily to be turned into presence-absence data, too), even at only relatively coarse numerical scales, because the benefits are considerable (as stated by *Howard et al. 2014*). One thing that should be mentioned is that plenty of abundance data and models did not perform well and, abundance was extremely difficult to predict (*Oppel et al., 2012*). Finding the underlying causes that influence abundance model accuracy and constructing more accurate models would be extremely important and useful in future applications toward individual-based policy applications.

For a spatial priority protection of mobile species, one should note that high numbers of individuals are not always present in the same habitats and pixels; instead, low numbers of individuals may occur in one place many times. In addition, this may have implications on spatial priority protection for mobile species. Previous studies have used analytical approaches to address some of these challenges (e.g., *Nichols, Thomas & Conn, 2009*; *Kery & Andrew, 2010*; *Oppel et al., 2012*; *Jiguet et al., 2013*). However, no general modeling framework has been proposed for dealing with all these analytical challenges simultaneously. This is exactly where our PI offers progress. We also thought the situation of mobile species selecting habitats could be divided into five scenarios: higher numbers and multi-frequency, higher numbers and lower frequency, low numbers and multi-frequency, low numbers and low frequency, and none. When a conservation plan is designed for a species, one should consider not only occurrence index and frequency but also abundance. Here, we proposed the priority protection index (PI; Eq. (1) and Fig. 4), based on the distribution of occurrence and abundance patterns, as a helpful tool for more quickly designs a priority protection plan compared to indices, and it is only based on distribution of occurrence or abundance.

To date, quantitative estimates of population size during global and local changes have actually proven to be difficult to forecast (*O'Grady et al., 2004*). This is a major hindrance for effective management, as population size and trends are considered among the best correlates of extinction risk (*O'Grady et al., 2004*). Such measures are commonly used in determining the conservation status of a species (e.g., International Union for Conservation of Nature (IUCN)). We argue that habitat loss remains the one and only powerful metric that can be obtained quickly on a landscape-scale in the absence of proper trend and abundance data (e.g., *Drew, Wiersma & Huettmann, 2011*). The relationship between predicted environmental suitability and abundance—as presented here—may provide us with a possible method for predicting population size and its changes associated with distributional changes; additionally, it may be particularly appropriate for non-mobile species (e.g., plants, fungi). However, this method is not particularly suitable for mobile species, especially for highly mobile species, such as many birds, bats, and flying insects. They may move over a large landscape within a single day, and abundance and the environment conditions can vary seasonally and spatially. When computing population

size or population density using abundance, the primary task will be determining the appropriate unit area for investigation and conservation management.

This study is the first that has combined model-predicted occurrence (representing species distribution models) and abundance indices (representing species abundance models) to produce a priority protection index (PI), which may contribute to spatial conservation and management decisions worldwide. We strongly encourage other researchers to test, apply and update the priority protection index (PI) to explore the generality of these findings further.

## ACKNOWLEDGEMENTS

We thank Min Liu for his hard work in the field, Jianguo Fu's great bustard photograph. Thanks also to a shared field survey among the authors. Further thanks go to Salford Systems Ltd. for providing the SPM software.

### Funding

This research is funded by the National Natural Science Foundation of China (project no. 31570532), the State Forestry Administration of China and the Scientific Research Committee of the China Wildlife Conservation Association (project no. kkw-2017-005). The funders had no role in study design, data collection and analysis, decision to publish, or preparation of the manuscript.

### Grant Disclosures

The following grant information was disclosed by the authors:
National Natural Science Foundation of China: 31570532.
State Forestry Administration of China.
Scientific Research Committee of the China Wildlife Conservation Association: kkw-2017-005.

### Competing Interests

The authors declare there are no competing interests.

### Author Contributions

- Chunrong Mi conceived and designed the experiments, performed the experiments, analyzed the data, contributed reagents/materials/analysis tools, wrote the paper, prepared figures and/or tables, reviewed drafts of the paper.
- Falk Huettmann analyzed the data, contributed reagents/materials/analysis tools, reviewed drafts of the paper.
- Rui Sun contributed reagents/materials/analysis tools, reviewed drafts of the paper.
- Yumin Guo reviewed drafts of the paper.

### Data Availability

The raw data has been provided as a Supplemental File.

## Supplemental Information

Supplemental information for this article can be found online at http://dx.doi.org/10.7717/peerj.4160#supplemental-information.

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
