# Peer review of "Combining occurrence and abundance distribution models for the conservation of the Great Bustard"

_PeerJ, doi:10.7717/peerj.4160_

## Round 0.1 · original submission · Major Revisions

As you can see, both reviewers require substantial numbers of changes to this version of the manuscript. Both comment that you need to pay closer attention to the English. Now, as someone who has many Chinese collaborators I know how difficult this can be. My best suggestion is that you consider the program Grammarly (www.grammarly.com). It's something I use routinely even on my writing.

And, of course, please address each point raised by the reviewers in a document detailing your changes.

Reviewer 1 ·

Basic reporting

The study forecasted the spatial occurrence as well as the abundance distribution of Great Bustard based on the priority protection index (PI) by combining occurrence and abundance indices. The results show that the occurrence and abundance models display different spatial distribution patterns, and PI could guide the protection of the areas with high occurrence and high abundance. Generally, the MS did not tell me how to combine the occurrence and abundance indices, and display clearly spatial distribution pattern. Furthermore, only based on the distribution data of Great Bustard in Bohai Bay, the author confirmed that the findings have a wide relevance and applicability, worldwide, and it was difficult to be a global research template. This is strongly non-robust.
In Introduction part, the author need to clearly show why many scholars found species occurrence and abundance distribution not to display similar patterns, conservation decisions based on SDMs predictions are insufficient and may even be misleading? And why one time-critical challenge and associated progress will be centered how to combine the useful information that SDMs and SAMs? And one of the objects of the MS was to assess and develop models to predict accurately the patterns of bustard occurrence and abundance. But in Method part, you just used the Random Forest model, no any information about the developing model, if you developed the model, please give detailed information of developing, if not, please describe your method accurately.

Experimental design

I don’t know how you choose the eleven environmental variable? Why these variables, not other variable? And why eleven, not ten or eight, or other? This need more details.
About the experimental design, you said “we travelled with a small four-wheel-drive tractor along fixed routes”, can you give the detailed information about the route line design context. Actually, this is key for your data robustness.

Validity of the findings

Results are not well stated linked to original research question. In Introduction part, you mentioned the second object of the study was to predict accurately the patterns of bustard occurrence and abundance, but in Result part, you did not give clearly the pattern? Why? I suggest you describe your predicted pattern in the Result part.

Additional comments

Line 25, best-available presence and count records for an endangered farmland species, Great Bustard (Otis tarda dybowskii) in Bohai Bay, China, as a case study. How did you proved that presence and count records for an endangered farmland species was best-available? And line28, how you know the method was a powerful machine learning method? Can you give some illustration?
Line 30-31, You mentioned that the environmental variables influenced bustard occurrence and abundance differently. How the variables influenced? And you found that occurrence and abundance models display different spatial distribution patterns. Please give actually pattern, what pattern would be displayed?
Line 36, only by your case study, how to confirmed that your findings have a wide relevance and applicability, worldwide. That means how to extend you scale from the local Bohai bay to worldwide scale?
Line 91, The topographical and climate condition varies little in the study area. Can you give information about how varies little?
Line 99, According to my knowledge, the bustard distributed widely, why you said Bohai bay is the world’s largest wintering ground ? Is it based on your research or other people’ research? And why is representative of the typical farmland situation in the North China Plain. Can you give actuately base or literature?
Line 123-124, please give more detailed information about how to construct a distance layer for these variables using the Euclidean.
Line 135-136, Please give detailed information that what range of NDVI representing good condition vegetation, and what range of NDVI represent the bed condition of vegetation, and why?
In line 139-155, there are many expression need to be confirmed, such as advanced (line 139), robust (line 141), very good (line 142), performed the best (line 144), robustness (line 146), not so good (line 148), better (line 151), best (line 153). All of these evaluation were based on your own judgment but not citation or proof. I suggested you should rewrite this part.
Line 47, you constructed initial abundance model, but you did not tell us how to construct the model?
Line 158, you adjusted the simulation abundance, can you give more information about how to adjusted the simulation?
Line 171, you extracted the habitat information, in line 172 you created a model file, and in line you adjusted the predicted abundance, can you tell us how to extract, create, and adjust?
Line 182, what is better?
Line 195, what is the more suitable and scientific? What is not suitable or scientific?
Line 258, how do you know the habitat had a fragmented distribution?
In Discussion part, some sentence is no cited literature, such as line 264, 366-367, and more. Please check.

Reviewer 2 ·

Basic reporting

This research looks at an endangered species and proposes a way for better conservation management. In the manuscript, the authors have done a good job to illustrate their design, data collection, methods for modelling, discuss the results and future implications. The references are thorough and figures are informative.

Experimental design

The design of the study is generally clear. However, the authors need to address the following questions. First, what is the reason and reference to use a linear regression to convert observation abundance to another? You just want better prediction or other reasons? Why not transform the abundance data before according to its distribution (zero-inflated)? Second, why choose random sampling for pseudo-absence? Do you control for the distance between presence and the random absences? Third, how do you validate your model? Do you have other independent dataset to validate it? Fourth, for this mobile species, for the same survey, how do you avoid counting the same individuals in different sites?

Validity of the findings

The findings are useful in conservation planning, especially the way to combine two different kinds of information: presence and abundance. The author could illustrate this findings more, what is the consequences of just using species distribution model or abundance model compared to using the combined result? You can threshold the two maps and compare the differences.

Additional comments

It is a good study. I have put other comments for the wording of this paper. Please address this issues and work on the language more carefully.

Line 35: spell out the rel.
Line 73: Delete “Farmland”. This species habitat contains both grassland and farmland.
Line 101: delete “situation”
Line 105: delete “winter survey”
Line 115: How do you avoid record same individuals during the survey?
Line 125: change spacing to pixel size.
Line 156: Is there any reference for this method (using linear regression converting observation to simulation abundance)?
Line 166: There are various ways to choose pseudo-absence. Why use random sampling? Do you control the distance between absence and presence?
Line 210: rephrase the sentence. Not clear what you want to say.
Line 215: rephrase this sentence.
Line 220: What is multivariate package? Not sure what it refers to
Line 263: change “!” to “.” Not common to use “!” in academic paper.
Line 276: Please rephrase this sentence. Can’t understand what you mean.
Line 309: You jumped around in this section. The author talked about the birds then jumped to plant and crop later. Please make clear transition.

Figure2. Change the x axis title for b. Is it observation ID?

External reviews were received for this submission. These reviews were used by the Editor when they made their decision, and can be downloaded below.

---

## Round 0.2 · accepted · Accept

Thank you for taking care of the reviewer's concerns. As you can see, you have satisfied all of them.

Reviewer 1 ·

Basic reporting

The revised MS solved the issues I have proposed.

Experimental design

The authors have given detailed informations about the chose of the eleven environmental variable.

Validity of the findings

The conclusions have been stated appropriately.

Additional comments

The authors have replied my questiones detailed.